# Model of multidisciplinary teamwork in hip fracture care: a qualitative interview study

Sarah Drew ,[1] Fiona Fox ,[2] Celia L Gregson ,[1] Rachael Gooberman-Hill [2,3]

¹Musculoskeletal Research Unit, Bristol Medical School, University of Bristol, Bristol, UK
²Bristol Medical School, University of Bristol, Bristol, UK
³UK National Institute for Health Research Bristol Biomedical Research Centre, University Hospitals Bristol and Weston NHS Foundation Trust and the University of Bristol, Bristol, UK

**Correspondence to**
Dr Sarah Drew;
sarah.drew@bristol.ac.uk

## ABSTRACT

**Objectives** Hip fractures are common injuries in older age with high mortality requiring multidisciplinary clinical care. Despite guidance, there is considerable variation in hip fracture services and patient outcomes; furthermore, little is known about how successful multidisciplinary working can be enabled. This study aimed to characterise professionals' views about the core components of multidisciplinary teamwork in hip fracture care.

**Design** The study comprised qualitative interviews with healthcare professionals delivering hip fracture care. Interviews were audio recorded, transcribed and analysed abductively: material was coded inductively and grouped into higher level concepts informed by theories and frameworks relating to teamwork.

**Setting** Four hospitals in England.

**Participants** Participants were 40 healthcare professionals including orthopaedic surgeons, orthogeriatricians, physiotherapists and service managers.

**Results** Results identified four components of successful multidisciplinary teamwork: (1) defined roles and responsibilities, (2) information transfer processes, (3) shared goals and (4) collaborative leadership. These were underpinned by a single concept: shared responsibility. Defined roles and responsibilities were promoted through formal care pathways, reinforced through induction and training with clear job plans outlining tasks. Information transfer processes facilitated timely information exchange to appropriate individuals. Well-defined common purpose was hindered by complex interdisciplinary professional relationships, particularly between orthogeriatric and orthopaedic staff, and encouraged through multidisciplinary team meetings and training. Clinical service leads were integral to bridging interdisciplinary boundaries. Mutual trust and respect were based on recognition of the value of different professional groups. Teamwork depended on formal clinical leads with facilitative and motivational roles, and on hospital leadership that created an environment supporting collaboration. Shared responsibility for patients was encouraged by joint orthopaedic and orthogeriatric care models. Staff shared responsibility by assisting colleagues when needed.

**Conclusions** Shared responsibility across the multidisciplinary team is fundamental to delivery of hip fracture care. Findings will inform development of clinical practice recommendations and training to build teamworking competencies.

## STRENGTHS AND LIMITATIONS OF THIS STUDY

⇒ Qualitative study design and methods using an abductive approach provided an in-depth understanding of care processes from the viewpoints of involved professionals and enabled the research to be informed by existing frameworks about teamwork.

⇒ The sample, comprising 40 participants drawn from four UK hospitals, achieved information power through inclusion of a range of professionals with relevant experience and with sufficient data to address the research aim.

⇒ Despite achieving information power, there are some professionals involved in healthcare and related discharge planning—such as social workers—who were not included in the study, meaning that we were unable to include their views.

⇒ Data collection was conducted in hospitals that admitted moderate to large numbers of patients with hip fracture and this may limit transferability of findings to smaller units.

## INTRODUCTION

Globally, hip fractures pose a major challenge to patients and healthcare systems, with 4.5 million such fractures expected to occur annually by 2050.[1] Hip fractures are most commonly sustained by older individuals and can be devastating with considerable impact on quality of life and a high risk of subsequent death.[2 3] Provision of hip fracture care is costly to healthcare providers, incurring $5.96 billion in direct medical costs each year in the USA and £1.2 billion in the UK.[4] Comprehensive multidisciplinary hip fracture care pathways reduce postoperative complications and mortality.[5] To make recommendations about how to further enhance the quality of care for people who have had hip fracture, there is a need to understand how healthcare professionals implement care pathways.

In the UK, thorough guidance has been published on the management of hip fracture.[6–8] Despite this, considerable variation

exists in the organisation of care pathways[7] [9] and in patient outcomes.[10] Hip fracture care requires communication and coordination across a range of specialties and professional roles including emergency medicine, surgery, anaesthetics, geriatric medicine and rehabilitation.[6] [7] This multidisciplinary care reflects patients' complex needs that may include multiple pre-existing conditions.[11] Guidelines recommend prompt care, with rapid transfers of care between professionals.[7] [11]

Studies about provision of acute rehabilitation for hip fracture with clinical leaders and physiotherapists have highlighted the importance of communication between professionals and patients in rehabilitation for hip fracture.[12] [13] Similarly, a recent interview-based study by Guerra and colleagues with healthcare professionals explored acute rehabilitation and implementation.[14] The study found that communication and collaboration are central to provision of effective rehabilitation and that organisational constraints are seen as a barrier to such provision. Other studies have also explored how patients and carers engage with hip fracture care, and qualitative findings highlight the emotional and practical challenges associated with hip fracture.[6–8] Together, these studies highlight the importance of understanding the way in which care for hip fracture is organised and delivered.

High-quality multidisciplinary care is crucial to delivery of safe and effective services. Multidisciplinary teamwork is necessary in delivery of clinical care for a number of health conditions and specialties, such as cancer,[15] dementia[16] and in older people's services.[17] 'Teamwork' can be defined as 'a set of inter-related thoughts, actions, and feelings of each team member that are needed to function as a team and that combine to facilitate coordinated, adaptive performance and task objectives resulting in value-added outcomes'.[18] Collaborative teamwork exists on a continuum of professional autonomy and integrative working. Three broad models describe collaborative teamwork: 'multidisciplinary', whereby multiple specialties with separate roles work within their own disciplines; 'interdisciplinary' that involves working together to deliver shared goals with some blurring of disciplinary boundaries; and 'transdisciplinary' where professional roles are shared.[19] Research has highlighted how different tasks across clinical disciplines shape how teams are organised and work together.[20]

Existing theories and frameworks guide professionals in the organisation and evaluation of the processes used in multidisciplinary team (MDT) communications, training and leadership. Some frameworks are generalisable across several areas,[18] [20–22] others have been developed for specific clinical disciplines.[23] Frameworks have identified diverse factors that promote clinical teamwork, including a shared understanding of team goals,[18] [19] [21] successful information exchange,[18] effective leadership[17] [18] [21] [23] and adaptability of services to meet changing demands.[18] [24] [25] Ellis and Sevdalis build on frameworks to identify core technical and non-technical skills to facilitate effective multidisciplinary teamwork in geriatric medicine.[17]

Non-technical skills include social, cognitive and personal resource skills. Strategic domains to enhance multidisciplinary working are skills (eg, good leadership), processes (eg, good documentation governance) and values (eg, prioritising patient needs and respect for colleagues). Evidence from behavioural research into multidisciplinary teamwork suggests that if any of these three components is missing, teamwork will be unsuccessful.[17]

Given the importance of MDTs in hip fracture care and previous learning from other fields, we aimed to develop a model of the core components needed to deliver successful multidisciplinary teamwork in hip fracture care in England. Understanding views about successful teamwork in the hip fracture care pathway will help provide visibility to core components and inform future work to support and enhance multidisciplinary care for hip fracture.

## METHODS

We used qualitative interviews to collect and interpret insights from health professionals engaged in delivery of hip fracture care in hospitals in England.

### Sample

#### Hospital sites

Healthcare professionals involved in the organisation and delivery of hip fracture were sampled from four National Health Service (NHS) hospitals across England. The NHS provides almost all hip fracture care in England, free to patients at the point of receipt.

Purposive sampling was used to identify study sites,[26] with variation in geography, number of hip fracture presentations per year, service configuration and audited hip fracture outcomes.[27] Three hospitals were urban and one rural. One hospital admitted a large number (>75th percentile), and three moderate (between 25th and 75th percentiles) numbers of patients with hip fracture annually. Hospitals are identified by pseudonyms to retain anonymity ('Springhill', 'Radford', 'Maplegrove' and 'Newbridge').

#### Participants

Forty interviews were conducted with healthcare professionals across the four hospitals. Staff were purposively sampled using a criterion approach to include healthcare professionals from across the care pathway, including orthopaedic surgeons, anaesthetists, orthogeriatricians, physiotherapists and occupational therapists, nurses, trauma coordinators and service managers.[28]

Potential participants were identified by a designated contact working within the service at each hospital site. Staff were invited to participate in the study by email, followed-up with reminder emails a minimum of 2 weeks later. Snowball sampling was also used,[29] with participants recommending other professionals involved in service delivery. In total, 75 healthcare professionals were approached to take part in the study, of whom 40

(53%) agreed to take participate. The remainder either declined or were unavailable. The final sample enabled achievement of information power as it was adequate to address the study aim through inclusion of appropriate participants across the care pathway.[30]

### Data collection

Two qualitative researchers (FF and SD) conducted interviews by telephone or teleconferencing technology (eg, Microsoft Teams), necessitated by COVID-19 pandemic restrictions. Interviews, lasting 45–90 min, were guided by a topic guide based on domains agreed collaboratively with healthcare professionals involved in delivery and organisation of hip fracture services. Part 1 focused on barriers and facilitators to the implementation of care. Part 2 covered multidisciplinary communication and cooperation including questions about access to resources, monitoring and evaluation (online supplemental material 1). Interviews were flexible to explore emerging ideas[31] and questions relating to teamwork were covered in all domains.

### Data analysis

Interviews were audio recorded, transcribed and anonymised to remove any identifying information. Participants were assigned pseudonyms. Data were analysed using an abductive approach.[32] Abduction is an approach that involves using research findings to develop new theories that build on existing theories and frameworks.[33] Further detailed explanation of abduction can be found in Tavory and Timmermans.[32] There are several stages to the abductive process. First, transcripts were imported into NVivo qualitative data management software[34] and coded using an inductive thematic approach.[35] In this process, we focused on aspects of the data relating to teamwork, which we found in all interviews. Next, codes and coded material were grouped into higher level concepts. In keeping with an abductive approach, the concepts were informed by existing frameworks about teamwork.[17 18 21] A model was then constructed to illustrate how core concepts contributed to multidisciplinary teamwork for hip fracture care (see figure 1). During this process all transcripts were analysed in the same way with the same staged process applied. To ensure rigour, 40% transcripts were independently analysed in duplicate (by SD and FF) and themes reviewed and refined to reach an agreed code list.[35]

### Patient and public involvement

To refine the study design and data collection materials, we held three meetings with members of 'The Patient Experience Partnership in Research' group, a dedicated musculoskeletal patient involvement group.[36] Members' input helped shape study documentation, and they felt that findings resonated with their own experiences, particularly reflecting their concerns that patient information was sometimes not well communicated within the system.

### RESULTS

The 40 interviewees comprised six orthopaedic surgeons; six anaesthetists; seven orthogeriatricians; three physiotherapists; four occupational therapists; one emergency medicine consultant and one emergency department nurse; three advanced nurse practitioners (ANP; in orthogeriatrics, emergency medicine and orthopaedics and trauma); one ward manager; one matron; two service managers; one senior operating department practitioner; one discharge coordinator; one trauma coordinator; one orthopaedic registrar; and one junior doctor. Between 5 and 15 participants took part from each hospital ('Springhill'=9, 'Radford'=11, 'Maplegrove'=15 and 'Newbridge'=5). Participants' characteristics are summarised by hospital to maintain anonymity (table 1).

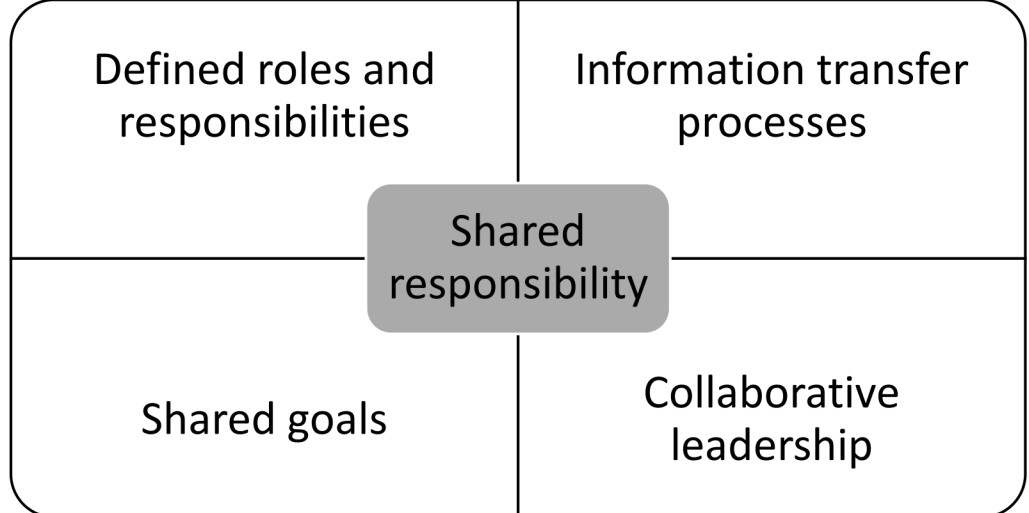

**Figure 1** Core components of successful multidisciplinary teamwork for hip fracture care.

**Table 1** Characteristics of healthcare professionals interviewed

| Professional role | Number of professionals interviewed across each hospital | | | | Time spent working in role (years) |
| | 01 Springhill | 02 Radford | 03 Maplegrove | 04 Newbridge | |
|---|---|---|---|---|---|
| Consultant geriatrician/orthogeriatrician | 3 | 2 | 1 | 1 | 3–18 |
| ED consultant | 1 | | | | 5 |
| Band 8 ANP emergency department | | 1 | | | 6 |
| Anaesthetist | 1 | 3 | 1 | 1 | 7–25 |
| Orthopaedic surgeon | | 2 | 4 | | 5–11 |
| Orthogeriatric advanced nurse specialist | 1 | | | | 14 |
| Physiotherapist | 1 | | 1 | 1 | 3–5 |
| Occupational therapist | 1 | 1 | 1 | 1 | 2–8 |
| MSK matron | 1 | | | | 7 |
| Service manager | | 1 | 1 | | 5 |
| Ward manager | | 1 | | | 5 |
| Senior theatre practitioner | | | 1 | | 8 |
| ANP | | | 2 | | 13–17 |
| Orthopaedic registrar | | | 1 | | 5 |
| FY2 doctor | | | 1 | | 3.5 |
| Discharge coordinator | | | 1 | | 19 |
| Trauma coordinator | | | | 1 | 5 |
| Total | 9 | 11 | 15 | 5 | |

ANP, advanced nurse practitioner; ED, emergency department; FY2, Foundation Year 2; MSK, musculoskeletal.

We identified four core components of successful multidisciplinary teamwork for hip fracture care and one central underpinning concept: (1) defined roles and responsibilities, (2) information transfer processes, (3) shared goals and (4) collaborative leadership. Across these, shared responsibility underpinned all activities (figure 1). We approach the results through a series of propositions with illustrative quotations shown in tables 2 and 3.

### Defined roles and responsibilities

*Proposition 1:* Successful multidisciplinary teamwork depends on clarity in roles and responsibilities within the team.

Clear roles and responsibilities were achieved by using protocols and formalised care pathways. These were undermined when key components of the pathway were missing from care provision; these included management of analgesia, optimisation for surgery and falls assessment. The extent to which protocols and pathway documents were used depended on 'buy in', which was variable, and partially related to staff seniority, with greater reluctance among most senior staff. Reasons for variation included disagreement about patient management and desire for greater professional autonomy. Participants provided examples where uncertainties and disagreement about responsibilities created difficulties. Comprehensive induction and training on protocols and pathways, particularly for new staff and rotating junior doctors, provided clear information on 'what to expect or adhere to'. Clarity was also achieved through job planning that outlined the parameters of professional roles and defined responsibilities.

### Information transfer processes

*Proposition 2:* Successful multidisciplinary teamwork depends on processes to facilitate the effective transfer and exchange of information to the 'right' individuals at the 'right' time. One of the primary objectives of care coordination was the effective transfer of information to inform patient management and decision-making. This was achieved via formal mechanisms including MDT meetings, ward rounds and documentation, shared information systems and informal and spontaneous communication.

MDT meetings in hospitals took a range of forms including trauma meetings of the trauma healthcare team, 'board rounds' (reviews of ward patients' progress in relation to discharge), 'huddles' (short briefings between team members) and discharge planning meetings (to plan for patients leaving hospital). Participants identified characteristics of successful MDT meetings, including establishing patient goals, regular attendance by a range of specialties and effective management such as time management and documentation of outputs. However, these were threatened

**Table 2** Core components of successful multidisciplinary working for hip fracture care: illustrative quotations

| Themes identified | Illustrative quotations (all names are pseudonyms) |
|---|---|
| **Defined roles and responsibilities** | |
| Proposition 1: Successful multidisciplinary teamwork depends on clarity in roles and responsibilities within the team. | |
| Effective use of hip fracture protocols and formalised care pathways. 'Buy in' of protocol and pathway documents. Comprehensive induction on protocols and pathways. Clear job planning to outline parameters of professional roles. | 'We have a good communication right from the start [of the pathway] … it's very kind of clear cut whose responsibility is what.' (Sue, Band 8 advanced majors' practitioner, emergency department, Radford) 'Not many people know what OT [occupational therapy] is and you've really got to fight sometimes about your professional role in things…. It's that you're just having to keep fighting, 'No, that's not for us. That's for somebody else'.' (Lois, trauma and orthopaedic ward occupational therapist, Newbridge) 'There was a complete lack of understanding from the trauma and orthopaedic consultants as to an expectation from them that orthogeriatrics was my biggest priority. And I was saying, well it's not in my job plan to do it and there was a complete lack of understanding from them about that.' (Elaine, consultant geriatrician, Newbridge) 'Well I think the protocols and pathways work really well if everybody adheres to it, but with having a large department with a variety of clinicians involved it's difficult to get a buy-in from everybody.' (Peter, consultant orthopaedic surgeon, Radford) 'Trying to get everybody to sign up to say something will always happen, seems to be quite difficult at the moment…. It does just take time and the people who have recently come through training tend to be far more cohesive and wanting to have demonstrable pathways that can be audited and guide people who are more junior.' (Tania, consultant anaesthetist, Newbridge) |
| **Information transfer processes** | |
| Proposition 2: Successful multidisciplinary teamwork depends on processes to facilitate the effective transfer and exchange of information to the 'right' individuals at the 'right' time. | |
| Use of a range of multidisciplinary team meeting forms including trauma meetings, focused 'board rounds', 'huddles' and discharge planning meetings. Effective multidisciplinary team meetings including establishing patient goals, regular attendance by a range of specialties and effective management. Accessible multidisciplinary team documentation that is completed consistently and appropriately designed. Shared information systems that cross organisational boundaries. Multidisciplinary team ward rounds to create shared knowledge and problem solve. Shared workspaces to promote verbal communication, including placing patients on a designated hip fracture ward. Spontaneous communication using technologies such as WhatsApp, mobile phone and email. Use of named coordinators as key points of contact, including trauma and discharge coordinators and hip fracture nurse specialists. Use of informal coordinators as points of contact. | 'I'm involved in the daily trauma meetings every morning, so that is all the consultants, that's the service manager, that's the junior doctors and that's us as an ACP [Advanced Clinical Practice] team. And you talk about the patients…. Sometimes the consultants can talk over you and forget that you're actually telling them what they need to know.' (Jacqui, advanced nurse practitioner, Maplegrove) 'I've got a bleep, they've got my mobile, so I'm contactable throughout the day. So people from wards will be ringing me to ask questions.' (Joanne, trauma coordinator, Newbridge) 'I think sometimes that probably when some of the consultants or the anaesthetists come on the ward that sometimes [ward staff are] a bit wary of approaching them.' (Barbara, advanced nurse practitioner, Maplegrove) [The trauma coordinator] role has sort of, it's found itself floundering a bit so we don't intend to remove it, we just intend to enhance it and make it more clear what the role [is]. (Darren, service manager, Radford) |
| **Shared goals** | |
| Proposition 3: Successful multidisciplinary teamwork depends on the existence of a shared and well-defined purpose within the team. | |

**Table 2** Continued

| Themes identified | Illustrative quotations (all names are pseudonyms) |
|---|---|
| Importance of dynamics between different professional groups. Impact of structural divisions within NHS Trusts that reinforce professional differences. Staff transience undermining shared goals. Use of daily multidisciplinary team meetings to build shared goals. Multidisciplinary team training to share knowledge and understanding. Importance of clinical leads in bridging professional boundaries. Communal efforts in monitoring and appraisal. | 'Essentially, I think we all are bound by sort of evidenced based [care], I think that is a common sort of thing and we all want the best for our patients. And I think those are two things which bind us that we all want to do the right thing for the patients.' (Dheeraj, consultant anaesthetist, Radford) <br> 'It's one of our major battles [as orthogeriatricians], the fact that we struggle to engage [with orthopaedic surgeons]. We struggle with them to engage with us, because you know basically they fix them and then walk away but occasionally we need them back and it's an ongoing battle, always has been.' (David, orthogeriatric advanced nurse specialist, Springhill) <br> 'I'm not a fan of the divisional structure of hospitals. I think it's quite artificial and hospitals are trying too hard to align themselves like a corporation would… I think it can [impede communication].' (Andy, consultant orthopaedic surgeon, Maplegrove) <br> 'Our top consultant that works on [the hip fracture] ward, she's very driven by being the best, and having the best outcome for our patients. So I think everyone has a bit of a collective drive, but I think the pathway, being on that ward and on the pathway is a mega drive. Because it's constantly evaluated.' (Jane, occupational therapist, Springhill) <br> 'Updating your team is massively important. If you explain to your team why you're doing something, what you'd like done and why you'd like it done, that really helps. It's just that you've got that shared aim at the end of it. If you're working towards the shared aim, I think things work much better.' (Lucy, emergency medicine consultant, Springhill) |
| **Collaborative leadership** <br> Proposition 4: Successful multidisciplinary teamwork depends on effective, collaborative leadership across several specialties and levels. | |
| Importance of nominated departmental leads for hip fracture care. Formal model of leadership in small, mixed specialty teams. Clinical service leads with a facilitative and motivational role. Use of multidisciplinary team planning meetings. Importance of leadership development and succession planning to maintain stability. Centralised leadership providing sufficient capacity and resources for hip fracture care. Support for service improvement initiatives and innovation. Designated hospital board executives who engage with clinical service leads and champion hip fracture care. | 'I mean I think there is good engagement in the service at a senior level, and for that reason I think that focus, that sort of filters down across other members of the team.' (Kate, consultant geriatrician, Springhill) <br> 'I think if you get a core of people who are interested and enthusiastic, and I think that's across all disciplines, then there's no doubt that commitment can then drive improvements in other areas.' (Jon, consultant anaesthetist, Radford) <br> 'You've got to find key people [from a range of specialties] that are passionate about [hip fracture care] and get it. And even understand that, you know, the frailest of the frail, and you've got to do it properly and that takes lots of different services and lots of different professionals to pull that together … I guess I've been lucky here because people have put their head above the parapet rather than me pulling them in sort of kicking and screaming.' (Amanda, consultant orthogeriatrician, Springhill) <br> 'So, I think what worked well initially was a lot of buy-in and support from management. So three, four years ago one of the senior business unit managers, called in a couple of consultants that said, 'I'll support whatever you want to help this pathway better, what can we do about it?'… I think [it's helpful because] it brings the whole team together.' (Peter, consultant orthopaedic surgeon, Radford) |

NHS, National Health Service.

by lack of attendance from key professional groups due to competing commitments and a view that attendance was 'a waste of time' as time would be better spent with patients, or because meetings often repeated issues without resolution.

MDT documentation was vital as a means to transfer patient information. However, patient documentation was not always accessible to relevant teams at key points in the care pathway, nor appropriately designed or consistently completed to capture all relevant information. Information transfer was facilitated through patient tracking systems, 'ward boards' displaying key patient information and electronic patient records. Communication processes were undermined when information technology systems did not cross boundaries either within secondary care, or with external organisations such as primary care and community services, including when primary care records could not be accessed.

**Table 3** Underpinning concept of shared responsibility: illustrative quotations

| Themes identified | Illustrative quotations (all names are pseudonyms) |
|---|---|
| **Proposition 5: Successful multidisciplinary teamwork depends on a culture of shared responsibility for care delivery.** | |
| Use of joint care models between orthopaedic surgeons and orthogeriatricians. Assistance from colleagues in performing tasks when needed. Provision of feedback to improve performance. Blurring of traditional role boundaries. Mutual trust and respect for one another's skills. Recognition of the value of different professional groups. Familiarity with colleagues to build trust. | 'When we're all talking together [as a team] sometimes we can go, 'Can you do this and we'll do that' and I suppose it's just supporting each other…. So again, sometimes it's just like, 'If you do this then that gives me time to do that' and we work that way.' (Jessica, occupational therapist, Radford) 'You've got to listen to everybody and without certain people in that team it won't work, so I form a good rapport with the social workers, placement of care hub, so that you know they can trust me and I can trust them.' (Alice, Foundation Year 2 doctor, Maplegrove) 'I think you need to appreciate each other in terms of what people bring. You need to recognise other people's strengths and whether they're best placed to talk about something or you are yourself.' (Andy, consultant orthopaedic surgeon, Maplegrove) 'I think there has to be an element of respect [for colleagues from different teams] as well, that in the good old days, a consultant was king and the staff were just, you know the minions, working minions, working around them but nowadays there's more of an equality in the relationship. So, I would probably quite happily give an opinion [to] the Nurse on the ward, if she has a query about a patient, cause she sees the patient every day, so there's quite an open communication both ways.' (Lance, consultant orthopaedic surgeon, Radford) 'At one point I did start going to the trauma meeting for a bit. And actually that was really interesting for me because I did realise, hearing them talk about how specialised their knowledge was actually about operations and the operative management of problems. And I found that really interesting because I thought gosh, you know, these people do know a lot about what they did.' (Elaine, consultant geriatrician, Newbridge) |

MDT ward rounds and 'huddles' provided opportunities to create shared knowledge and problem solve. Participants advocated varied modes of communication to bring together professional groups at different stages of the care pathway.

Shared workspaces promoted frequent and spontaneous verbal communication. Allocating patients to specific hip fracture wards facilitated this exchange as healthcare professionals were concentrated together. Spontaneous communication was also achieved using technologies such as WhatsApp, mobile phone and email. However, success depended on the interpersonal skills of staff. Verbal communication was hindered by hierarchical relationships. While some senior members of staff emphasised lack of hierarchy on wards, one ANP felt that ward staff were 'wary' of discussing patient management with consultants. They felt this was due to ward staff lacking confidence, coupled with the belief in persistence of traditional hierarchy. The ANP therefore acted as a go-between while also encouraging ward staff to approach and communicate directly with consultants. 'Coordinators' functioned as key points of contact, providing continuity and daily communication between professional groups at different stages of the care pathway. Formal, named coordinator roles included trauma and discharge coordinators and hip fracture nurse specialists. Informal coordinating roles provided teams with an additional point of contact. In one site this included a named member of staff with responsibility for patients with hip fracture. However, over-reliance on individuals meant that when they were away, communication and coordination was compromised.

### Shared goals

*Proposition 3:* Successful multidisciplinary teamwork depends on the existence of a shared and well-defined purpose within the team.

Shared goals were undermined by complex dynamics between different professional groups, particularly geriatric and orthopaedic staff whose participants felt had different priorities. Several orthogeriatricians described orthopaedics as a specialty that focused on 'fixing' the fracture, compared with their holistic approach. The structural divisions in NHS Trusts that placed staff within different 'directorates' were seen as reinforcing professional differences and impeding communication. Engagement with shared goals was also challenging due to staff transience, particularly of junior doctors.

Shared goals were encouraged through multiple processes. Daily MDT meetings enabled staff to build shared goals and collaborative treatment plans for individual patients and to hear 'different points of view'. MDT training provided opportunities for staff to share knowledge and understanding. This was particularly evident in the training, mentoring and support for junior staff. Clinical leads were integral to bridging professional boundaries and creating a common professional identity across individual specialties. They did this by establishing shared priorities and goals in MDT planning meetings, sharing monitoring and auditing results with colleagues

across the care pathway and using audit findings to drive service improvement. Communal efforts in monitoring and appraisal provided further opportunities to reinforce shared priorities and goals, such as reviewing National Hip Fracture Database data—a national audit programme to benchmark services, mortality events, internal audits and informal discussions about service improvement.

### Collaborative leadership

*Proposition 4:* Successful multidisciplinary teamwork depends on collaborative leadership across several specialties and levels.

Successful multidisciplinary teamwork relied on presence of nominated departmental leads for hip fracture care. These formal clinical leads had a facilitative and motivational role and centralised hospital leadership created an environment to support collaboration. Participants advocated a formal model of leadership that was concentrated in small, mixed specialty teams across orthopaedic, orthogeriatric, aesthetic and nursing divisions. Participants identified specialties missing at this level, including anaesthetists. Therapy staff felt excluded at this level and thought that their representation could promote more effective collaboration. Senior-level leadership was thought to promote commitment among colleagues at all levels.

Clinical service leads had a facilitative and motivational role in engaging healthcare professionals from a range of specialties and assembled teams to develop shared priorities and goals, joint protocols, audit priorities and mutually owned quality improvement plans. MDT planning meetings were a key part of this process. Participants identified several attributes that enabled 'heroic' individuals to fulfil these roles, such as 'passion' and 'drive'. The visibility of clinical service leads within the service and respect for their clinical judgement legitimised their position and influenced their success in achieving these aims. Hospitals with one strong clinical service lead emphasised the importance of leadership development and succession planning to maintain team stability and performance.

Effective leadership on the wards depended on centralised leadership that created an environment to support collaboration. Support included providing sufficient capacity and resources for hip fracture care including capacity for staff to attend MDT meetings, supporting local leads to engage in service planning and training by building this into job plans and supporting service improvement initiatives and innovation to help provide a sense of shared purpose. Some participants expressed frustration that reimbursement from best practice tariffs for hip fracture care was not sufficiently reinvested into service development. Participants emphasised the need for designated hospital board executives who engaged in bidirectional communication with clinical service leads and championed hip fracture care to facilitate this. Several participants highlighted a lack of centralised leadership due to the transience of higher management roles and competing priorities within the Trust.

### Underpinning concept: shared responsibility

*Proposition 5:* Roles and responsibilities, information transfer processes, shared goals and collaborative leadership all depend on a culture of shared responsibility for care delivery (table 3).

Underpinning and crossing all components, shared responsibility was evident and articulated. A culture of shared responsibility for hip fracture care ensured all members of the MDT were invested in care delivery. This was enabled through care models where patients were cared for jointly by orthopaedic surgeons and orthogeriatricians. Staff valued the assistance of colleagues in performing tasks when needed, particularly when they were overloaded. Shared responsibility was also evidenced by colleagues providing feedback to improve performance and taking on supervisory roles. Orthogeriatricians were seen as occupying a key role in supervision of and support for ANPs, therapists and junior doctors (medical and surgical). There was evidence that traditional role boundaries were blurred. This included the involvement of nurses in mobilisation of patients when therapists were not available, and anaesthetists' work to upskill emergency department doctors in nerve block provision.

Mutual trust and respect for one another's skills created an environment that encouraged staff to participate actively in the MDT including in meetings, discussions with colleagues and constructive management of disagreements. Trust and respect were based on a recognition of the value of different professional groups. Familiarity with colleagues, both professionally and personally, helped build this trust. Informal relationships were facilitated through shared workspaces. Mutual trust and respect were often cultivated over time.

### DISCUSSION

This study has characterised how hospitals deliver multidisciplinary work for hip fracture care and suggests core components needed to deliver this. Findings identified four components to promote successful multidisciplinary teamwork: (1) defined roles and responsibilities, (2) information transfer processes, (3) shared goals and (4) collaborative leadership. Shared responsibility underpinned and acted across all four components.

### Relationship to current literature

This study complements existing work on optimal care models for hip fracture care by exploring how multidisciplinary teamwork can be organised and delivered in practice. Guidelines and existing research have highlighted the importance of implementing formal processes to coordinate care, including use of well-defined care pathways and protocols[5 37] and good communication to facilitate information transfer.[9] The alignment of team goals

through multidisciplinary and clinical governance meetings and shared learning has also been identified within the acute rehabilitation of patients with hip fracture. The National Institute for Health and Care Excellence implementation advice for hip fracture management and British Orthopaedic Association guidelines have highlighted the importance of identifying an overarching clinical lead for hip fracture care, supported by senior leads from a range of specialties.[9] Coordination between orthogeriatricians and orthopaedic surgeons[38 39] and formal roles such as trauma and discharge coordinators have been highlighted elsewhere as important.[40–42]

Our findings also highlight possible barriers to effective multidisciplinary teamwork. These included some reluctance by more experienced staff to use hip fracture pathway documents or newly developed protocols, as well as barriers to communication presented by belief in hierarchy. Findings reflect wider systemic issues that have been identified in the context of other MDTs in the NHS. These include the existence of structural arrangements within and between secondary care and external organisations that hinder communication, such as incompatible computing systems,[43] staff transience[17] and steep hierarchies that stymie communication within teams.[44] The socialisation of staff into different professional groups with distinct cultures, identities and priorities has further been identified as a hindrance to collaborative work.[45] The model of clinical service leadership described here reflects a broader trend in healthcare away from a 'command and control' approach to a motivational and facilitative style that involves building relationships with staff from a range of specialties.[46] It also reflects increasing recognition of the value of collaborative leadership in which leadership roles are shared between team members.[47]

Hip fracture care includes separate roles with some evidence of interdisciplinary working through blurring of traditional role boundaries.[19] Core components of teamwork in this context encompass a range of skills (eg, leadership), processes (eg, hip fracture protocols and pathways, formal mechanisms for information transfer) and values (eg, shared goals).[17] Several strategies to support multidisciplinary teamwork are reflected in previous work, including interdisciplinary training,[48] shared monitoring and evaluation processes[17] and familiarisation with colleagues to build trust.[21]

Our study has identified the centrality of shared responsibility within MDTs delivering care for hip fracture. Care processes such as MDT meetings, monitoring and auditing processes and MDT training can serve dual purposes by creating and sustaining shared responsibility as a value.

This work is taking place within a broader programme to develop a toolkit to enable service providers and commissioners to improve organisation and delivery of hip fracture services,[27] results due in 2023. Based on our findings, we have developed our model into a how-to guide of practical recommendations for hip fracture care,

including interdisciplinary 'exchange' training built into both orthopaedics and orthogeriatrics specialty training curricula (online supplemental material 2). This resonates with work focused on other conditions, including cancer, that require multidisciplinary teamwork.[22 49]

## Strengths and limitations

The sample, comprising 40 participants drawn from four UK hospitals, provided sufficient data to address the research area and included professionals with relevant experience, as such the study achieved information power.[30] Using an abductive analysis approach, rather than transposing data onto an existing theory or framework, ensured the results were data driven such that data were not forced into predefined constructs.[34] Accordingly, where relevant, data were coded into more than one concept. We invited all healthcare professionals who are working within the service to participate. However, it is possible that those who declined were less engaged in service delivery; we also note that others involved in care or discharge planning—such as social workers—did not take part and that numbers of allied health professionals (7 out of 40) were relatively low compared with their overall representation in trauma care. Despite these limitations, the study included participants from diverse professional backgrounds, and the candid way that they spoke about challenges provided us with confidence in the veracity of the data. As data collection took place in hospitals that admitted moderate to large numbers of patients with hip fracture, the transferability of findings to smaller units may be limited and there may be need for more work to understand fracture care in such settings.[50]

## Further research

Further work is needed to understand leadership configurations in hip fracture care, particularly the role of 'emergent' leaders who do not have formal roles but exert considerable influence within teams.[51] Future research is also needed in smaller contexts and we are planning to evaluate the impact of our toolkit on the organisation and delivery of hip fracture services.

## CONCLUSIONS

This study used qualitative methods to develop a description of core components needed to deliver multidisciplinary teamwork for hip fracture care. It highlights the importance of creating and maintaining a culture that includes shared responsibility for care delivery. Our findings provide a basis for future training, skills development and assessment of multidisciplinary teamwork for hip fracture care in the UK.

**Acknowledgements** We thank all the healthcare professionals who participated in this study and the REDUCE study group who provided support and guidance. The REDUCE study group consists of Dr Rita Patel, Professor Andrew Judge, Professor Antony Johansen, Dr Elsa M R Marques, Dr Petra Baji, Ms Jill Griffin, Ms Marianne Bradshaw, Dr Katie Whale, Mr Tim Chesser, Professor Xavier L Griffin, Dr

Muhammad K Javaid and Professor Yoav Ben-Shlomo. We are grateful to members of the Patient Experience Partnership in Research (PEP-R), whose input shaped the research design and how it was carried out.

**Contributors** All authors contributed to the design of this research and the acquisition, analysis and/or interpretation of data. All authors contributed to drafting of this work and revising it for important intellectual content, and gave final approval for the version to be submitted. RG-H is responsible for the overall content as guarantor. The authors comprise two social scientists with several years of experience in health research (SD, FF), a practising consultant orthogeriatrician with academic expertise in epidemiology and population health research (CLG) and a social anthropologist with expertise in qualitative and interdisciplinary research (RG-H). We brought our different backgrounds into the research including its design and conduct, interpretation of the data and results.

**Funding** This work was supported by Versus Arthritis (reference number: 22086).

**Competing interests** None declared.

**Patient and public involvement** Patients and/or the public were involved in the design, or conduct, or reporting, or dissemination plans of this research. Refer to the Methods section for further details.

**Patient consent for publication** Not applicable.

**Ethics approval** This study involves human participants and was approved by the Faculty of Health Sciences University of Bristol Research Ethics Committee (reference number: 108284) and by the NHS Health Research Authority (20/HRA/71). Written informed consent was provided by all participants, confirming understanding of voluntary participation and potential publication of anonymised interview quotations. NHS settings (Trusts) for the four study sites each provided research governance approval. Participants gave informed consent to participate in the study before taking part.

**Provenance and peer review** Not commissioned; externally peer reviewed.

**Data availability statement** Data are available upon reasonable request. Anonymised interview data may be accessed via the University of Bristol Research Data Repository. Access to the data will be made available to researchers for ethically approved research projects, from a 6-year embargo period, which ends on 1 April 2029. Data can be accessed on the understanding that confidentiality will be maintained and after a Data Access Agreement has been signed by an institutional signatory. No authentic request for access will be refused. Data are available at the University of Bristol data repository, data.bris, at https://doi.org/10.5523/bris.3f9j9z626s7f32a3r5j1wd7zt7.

**ORCID iDs**
Sarah Drew http://orcid.org/0000-0002-2092-8506
Fiona Fox http://orcid.org/0000-0001-7313-8105
Celia L Gregson http://orcid.org/0000-0001-6414-0529
Rachael Gooberman-Hill http://orcid.org/0000-0003-3353-2882

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
