## [Reviewer comments · BMJ Open]

ARTICLE DETAILS

TITLE (PROVISIONAL)	A model of multidisciplinary teamwork in hip fracture care: a qualitative interview study
AUTHORS	Drew, Sarah; Fox, Fiona; Gregson, Celia; Gooberman-Hill, Rachael

VERSION 1 – REVIEW

REVIEWER	Snowdon, David Monash University, Peninsula Clinical School
REVIEW RETURNED	04-Jan-2023

GENERAL COMMENTS	Thank you for the opportunity to review this manuscript reporting on a qualitative study that investigated the core components of effective multidisciplinary teamwork in hip fracture care. I believe that this is a well-conducted qualitative study that addresses an important topic. I have included some points for clarification and suggestions on how to improve the manuscript below. Abstract: Page 4 line 12: missing word (? in) 'Hip fractures are common injuries older age...' Article Summary: Page 5 line 49: Only need to mention hip fracture once in following text 'This is the first study to understand and characterise the core components needed to deliver multidisciplinary teamwork for effective and efficient hip fracture care delivery for hip fracture.' Introduction: Page 7 line 33: You state 'However, research to date has not explored care delivery from the viewpoint of professionals.' However, delivery of care for people with hip fracture from the viewpoint of professionals has been explored in several studies (see below). The introduction should make mention of this existing literature and highlight the evidence gap with reference to these studies. Guerra et al. Multidisciplinary team healthcare professionals' perceptions of current and optimal acute rehabilitation, a hip fracture example A UK qualitative interview study informed by the Theoretical Domains Framework. PLoS One 2022. DOI: https://doi.org/10.1371/journal.pone.0277986 Hordvik et al. Physiotherapists' experiences with older adults' rehabilitation trajectory after hip fracture: A qualitative study in Western Norway. Physiotherapy Theory and Practice (in press): DOI: https://doi.org/10.1080/09593985.2021.2007557 Christie et al Improving the experience of hip fracture care: A multidisciplinary collaborative approach to implementing evidence-based, person-centred practice. International Journal of
--

	Orthopaedic and Trauma Nursing 2015. DOI: https://doi.org/10.1016/j.jjotn.2014.03.003 Taylor et al. Discharge planning for patients receiving rehabilitation after hip fracture: A qualitative analysis of physiotherapists' perceptions. Disabil Rehabil 2010. DOI: https://doi.org/10.3109/09638280903171568 Methods: Page 12 line 40: Please clarify how the remaining 60% of transcripts were analysed. Page 12: Please discuss research team reflexivity and how this was addressed. Results: Page 13, Table 1: In the main text you state that 3 physiotherapists participated in the study (Page 13, line 18) however in the study it states that 2 physiotherapists participated in the study. Please clarify. Page 16/17: You state 'While some senior members of staff emphasised lack of hierarchy on wards, one Advanced Nurse Practitioner felt that ward staff were "wary" of discussing patient management with consultants'. Were there any data that may shed some light on why this was the case? Discussion: In the results there are several findings that reveal possible barriers to effective multidisciplinary team work including: - Staff seniority (page 15 line 3 - The extent to which protocols and pathway documents were used depended upon 'buy in', which was variable, and partially related to staff seniority, with greater resistance from staff towards the end of their careers. - Hierarchy (Page 16/17 - 'While some senior members of staff emphasised lack of hierarchy on wards, one Advanced Nurse Practitioner felt that ward staff were "wary" of discussing patient management with consultants'.) - Exclusion of therapy staff from senior level leadership (page 24 line 30) These potential barriers are some of the more interesting findings from the study and a thorough discussion of these potential barriers would improve the discussion (particularly the relevance for clinicians and healthcare services who want to improve their hip fracture care) i.e. I believe that discussing these points would improve the clinical relevance of your manuscript. In the strengths and limitations section it would be worthwhile acknowledging the relatively small number of therapy (i.e. allied health) professionals. Also, there is no representation of some allied health professions who are less frequently involved in the care of people with hip fracture (e.g. dietitians and social workers) but nonetheless still have an important role in the management of some people with hip fracture. Representation from these professional groups who do not necessarily play a role in care of ALL people with hip fracture (compared to professions such as physiotherapists who mobilise the majority of people with hip fracture) may reveal other barriers and considerations to/for multidisciplinary teamwork (as opportunities for multidisciplinary communication may be less frequent with these professions).
--	--

REVIEWER	Tutton, Elizabeth
----------	-------------------

	University of Oxford, NDORMS
REVIEW RETURNED	11-Jan-2023

GENERAL COMMENTS	1) This is a well written paper about the key ingredients for multidisciplinary teamwork in hip fracture pathways. It provides important insights into staff views that will be useful for teams working in this area and will contribute to the broader aims of the main study. Its strength is that it is based on staff experience of their roles and what it is like to try to fulfil them when working in UK NHS Trusts. There are some aspects of this article that may be worth considering. 2) I found the line of argument, although sound, was not immediately clear. The abstract did not draw me in to read the paper. The five themes are clear but I think it lacked a central concept tying them together. It seemed to be teamwork, optimising collaborative teamwork, common goals/shared responsibility and in the discussion there is a 'new concept shared responsibility'. A central concept might provide a way of encapsulating the themes, frame the findings, make the themes clearer for the reader but also provide direction for future teamwork. I did wonder from reading your quotes whether shared responsibility and thus ways of working together was the glue that made all the processes/activities you identify work in practice. It is interesting that many of the quotes identify the daily struggles of working in a multidisciplinary team across multiple teams/departments. They do bring your work alive and make it more relatable. If you have a new concept, it needs to be clearly identified in the abstract and main text. This may help to identify what your study adds. 3) It would be useful to add a bit more about the principles underpinning the methodology and methods. Rigor notably needs to link with your methodology. The process of analysis was not totally clear to me. I understand the inductive element but needed more information on how it then linked to theory. For example, what was the content of these theoretical concepts and how did the findings fit or differ from the theoretical concepts? 4) In the findings the propositions were really helpful and very clear. You might consider providing a table with the components and their subcategories, their propositions and supporting quotes. This would provide a sense of the whole and visually link the quotes directly to the propositions. More quotes may also be useful. Some groups appeared underrepresented. Quotes can also be useful to show the overlaps between components. Figure 1 was not that useful due to the lack of content. There were a lot of barriers and some facilitators to each component. I wondered if a bullet point list in a box for each component might signpost the reader to key messages in the findings. I understand there will be a great deal of data for this study and it is hard work summarising it in 4,000 words. 5) Effective multidisciplinary teamwork is used and I would suggest that this needs qualifying to identify it as staff views about effectiveness or use another word, due to its links with quantitative research. 6) Explain a few terms in context, e.g board rounds, huddles. 7) In my view I would be cautious about identifying the paper as the first in a vast subject such as teamwork. Staff views, in the context of hip fracture, the current culture and at this particular time may be key. It is suggested that the model will be useful outside of the UK which does not take into account context and
---

	culture. I would be inclined to revisit the strengths and limitations section below the abstract and check that there is enough information to support these in the text. 8) Abstract, I would identify all the types of staff you included. 9) Did anything useful evolve from the PPI groups that influenced any part of the study? All the best for your ongoing work.
--	---

VERSION 1 – AUTHOR RESPONSE

Reviewer: 1 Dr. David Snowdon, Monash University, Peninsula Health

1. Comments to the Author:

Thank you for the opportunity to review this manuscript reporting on a qualitative study that investigated the core components of effective multidisciplinary teamwork in hip fracture care. I believe that this is a well-conducted qualitative study that addresses an important topic. I have included some points for clarification and suggestions on how to improve the manuscript below.

Thank you, we are pleased that you found our study to be well conducted and we appreciated your suggestions.

2. Abstract:

Page 4 line 12: missing word (? in) 'Hip fractures are common injuries older age...'

Thank you, we have added 'in' before 'older age' to correct this omission, we are sorry that you had to spot it in our behalf (page 3, line 42).

3. Article Summary:

Page 5 line 49: Only need to mention hip fracture once in following text 'This is the first study to understand and characterise the core components needed to deliver multidisciplinary teamwork for effective and efficient hip fracture care delivery for hip fracture.'

Thank you, we have edited the article summary to reflect the revised content and this sentence is now rewritten to explain that the study identified a common concept in core components p(age 4, lines 80-81).

4. Introduction:

Page 7 line 33: You state 'However, research to date has not explored care delivery from the viewpoint of professionals.' However, delivery of care for people with hip fracture from the viewpoint of professionals has been explored in several studies (see below). The introduction should make mention of this existing literature and highlight the evidence gap with reference to these studies.

Thank you for this enormously helpful comment and suggestion. We are sorry that we were overly general when we described the evidence gap. To address this, we have reshaped the introduction, and added reference to Guerra, Hordvik and Christie, including description of their key conclusions that relate to the importance of communication, collaboration and organisational barriers to effective acute rehabilitation. This now leads onto the discussion of previous work, including other clinical fields, about multidisciplinary teamwork, which provide ways of thinking about multidisciplinary teamwork. Our study explores teamwork in hip fracture care and as such contributes to broader literature about multidisciplinary work. These changes can be found in the introduction and we have removed some text and references to make space for the additions Thank you for this helpful comment and for sending suggested literature to us(page 6-7, lines 132-142, references 12, 13 and 14).

5. Methods:

Page 12 line 40: Please clarify how the remaining 60% of transcripts were analysed.

Thank you, we see that we had tried to be too brief in our description of the analysis and have explained the stages of the process more clearly and added explanation that all material was analysed in the same way (i.e. 100% analysed with the inductive process, 40% of which had a double coding process applied) This revision also addresses a comment made by the second reviewer (page 11, lines 240-254).

6. Methods:

Page 12: Please discuss research team reflexivity and how this was addressed.

Thank you, we find reflexivity in research enormously interesting. We bring social science and clinical experience to our research design and conduct. Interviews were conducted by individuals with social science backgrounds, and other members of the team were a social anthropologist and a consultant orthogeriatrician. As a team this enabled us to bring different perspectives—epistemological, methodological and field-related—into the work. We have added our disciplinary backgrounds into the author contribution statement, but have concerns that a full, appropriate discussion of reflexivity is beyond the bounds of this article, and we would be concerned about a brief addition failing to do justice to the matter or looking tokenistic. We hope that our solution is adequate, we appreciate that there are many ways in which reflexivity could be addressed Acknowledgements).

7. Results:

Page 13, Table 1: In the main text you state that 3 physiotherapists participated in the study (Page 13, line 18) however in the study it states that 2 physiotherapists participated in the study. Please clarify.

Thank you for highlighting this inconsistency. Three physiotherapists participated and so we have amended table 1 accordingly.

8. Results:

Page 16/17:

You state 'While some senior members of staff emphasised lack of hierarchy on wards, one Advanced Nurse Practitioner felt that ward staff were "wary" of discussing patient management with consultants'. Were there any data that may shed some light on why this was the case?

Thank you for this query and we agree that it is useful to add more detail about this issue, which arose in discussions of communication within multidisciplinary teams. We have therefore added text describing some information provided by an Advanced Nurse Practitioner (ANP) who thought that some ward staff did not feel confident approaching consultants because of their beliefs about hierarchy (pages 16-17, lines 351-356).

9. Discussion:

In the results there are several findings that reveal possible barriers to effective multidisciplinary team work including:

- Staff seniority (page 15 line 3 - The extent to which protocols and pathway documents were used depended upon 'buy in', which was variable, and partially related to staff seniority, with greater resistance from staff towards the end of their careers.
- Hierarchy (Page 16/17 - 'While some senior members of staff emphasised lack of hierarchy on wards, one Advanced Nurse Practitioner felt that ward staff were "wary" of discussing patient management with consultants'.)
- Exclusion of therapy staff from senior level leadership (page 24 line 30)

These potential barriers are some of the more interesting findings from the study and a thorough discussion of these potential barriers would improve the discussion (particularly the relevance for clinicians and healthcare services who want to improve their hip fracture care) i.e. I believe that discussing these points would improve the clinical relevance of your manuscript.

Thank you for this suggestion. We agree that it is important to include these barriers in the discussion and have therefore added text about the barriers that include some resistance to

use of pathway documents or protocols, and that hierarchy can inhibit communication. We have taken care only to include items contained in the results section and so have reduced other text relating to therapy staff, which is not reflected here. We have placed the new text about barriers next to the section about systemic issues found within the NHS in other multidisciplinary teams (changes throughout the Discussion).

In the strengths and limitations section it would be worthwhile acknowledging the relatively small number of therapy (i.e. allied health) professionals. Also, there is no representation of some allied health professions who are less frequently involved in the care of people with hip fracture (e.g. dietitians and social workers) but nonetheless still have an important role in the management of some people with hip fracture. Representation from these professional groups who do not necessarily play a role in care of ALL people with hip fracture (compared to professions such as physiotherapists who mobilise the majority of people with hip fracture) may reveal other barriers and considerations to/for multidisciplinary teamwork (as opportunities for multidisciplinary communication may be less frequent with these professions).

Thank you, it was helpful to know that we should include comment about allied health or similar professionals who were not included in the study. We have added brief information about staff not included (e.g. social workers) and the small numbers of allied health professionals. This is in the strengths and limitations section, this also now reflected in the article summary (page 5, lines 88-82; page 28, lines 537-540).

Reviewer: 1

Competing interests of Reviewer: I have no competing interests to declare.

Reviewer: 2 Dr. Elizabeth Tutton, University of Oxford

Comments to the Author:

1) This is a well written paper about the key ingredients for multidisciplinary teamwork in hip fracture pathways. It provides important insights into staff views that will be useful for teams working in this area and will contribute to the broader aims of the main study. Its strength is that it is based on staff experience of their roles and what it is like to try to fulfil them when working in UK NHS Trusts.

Thank you, we are pleased that you found our paper to be well written. We were keen to give voice to staff and to provide an accurate representation and conceptualisation of multidisciplinary teamwork in hip fracture care.

There are some aspects of this article that may be worth considering.

We have revised the manuscript in light of the helpful suggestions. To make additions while keeping the overall wordcount below 4,000 words we have removed and reduced text elsewhere throughout the manuscript. We trust that this is acceptable. All of the changes can be seen in track changes.

2) I found the line of argument, although sound, was not immediately clear. The abstract did not draw me in to read the paper. The five themes are clear but I think it lacked a central concept tying them together. It seemed to be teamwork, optimising collaborative teamwork, common goals/shared responsibility and in the discussion there is a 'new concept shared responsibility'. A central concept might provide a way of encapsulating the themes, frame the findings, make the themes clearer for the reader but also provide direction for future teamwork. I did wonder from reading your quotes whether shared responsibility and thus ways of working together was the glue that made all the processes/activities you identify work in practice. It is interesting that many of the quotes identify the daily struggles of working in a multidisciplinary team across multiple teams/departments. They do bring your work alive and make it more relatable. If you have a new concept, it needs to be clearly identified in the abstract and main text. This may help to identify what your study adds.

Thank you, the common thread is the new concept of 'shared responsibility'. In the submitted manuscript we had described this as one of the five themes, but we agree that it would be better to describe shared responsibility as a thread that flows through the entire work, and across all of the areas. We have made revisions throughout the manuscript so that shared responsibility is framed in this way, and we call it an underpinning 'concept'. Changes can be found throughout in the abstract, results and discussion. In some places we have moved text around to retain flow in these sections. We have also produced two tables of quotations, so that participant quotations relating to the four components and related propositions are on Table 2 and quotations relating to the single underpinning concept of shared responsibility are shown on Table 3.

3) It would be useful to add a bit more about the principles underpinning the methodology and methods. Rigor notably needs to link with your methodology. The process of analysis was not totally clear to me. I understand the inductive element but needed more information on how it then linked to theory. For example, what was the content of these theoretical concepts and how did the findings fit or differ from the theoretical concepts?

Thank you, we have made some changes to the description of the analysis approach, which we hope more clearly described the abductive approach and may enable readers to assess rigour in our research. Unfortunately, our description remains necessarily brief to remain within recommended wordcount and we have made careful removals throughout the manuscript to make space for all additions. We recognise this is not optimal and so we have also added clearer signposting to key literature about abduction (Tavory and Timmermans), should readers be interested. We are advocates for explicit use of abduction in qualitative health research, which we think is often not clear, perhaps there is another time and place for us to describe the value and process of abduction more clearly (page 11, lines 241-254).

4) In the findings the propositions were really helpful and very clear. You might consider providing a table with the components and their subcategories, their propositions and supporting quotes. This would provide a sense of the whole and visually link the quotes directly to the propositions. More

quotes may also be useful. Some groups appeared underrepresented. Quotes can also be useful to show the overlaps between components. Figure 1 was not that useful due to the lack of content. There were a lot of barriers and some facilitators to each component. I wondered if a bullet point list in a box for each component might signpost the reader to key messages in the findings. I understand there will be a great deal of data for this study and it is hard work summarising it in 4,000 words.

Thank you, this was very helpful indeed. We have revised the presentation of the quotations so that they are as tabulated as you suggest, with quotations listed under each component and with a separate table for the underpinning concept. We revised the figure to reflect the reframing of shared responsibility as the central theme and have also made edits in abstract and main content. The figure remains simple, and we hope it serves as an easy visual representation of use to readers, we would be happy to remove it if you think best (Figure 1, Tables 2 and 3).

5) Effective multidisciplinary teamwork is used and I would suggest that this needs qualifying to identify it as staff views about effectiveness or use another word, due to its links with quantitative research.

Thank you, this is enormously helpful. We were probably too close to our work to realise that this was needed and we have made additions throughout the text to make it clear that the study is about staff views of effectiveness and that the focus is on 'successful' teamwork. We hope that this improves clarity and helps readers to see the aims of our work more clearly. These changes and additions have been made throughout the manuscript including in the abstract and main content.

6) Explain a few terms in context, e.g. board rounds, huddles.

Thank you. We have added explanation of some terms where needed, including to board rounds, huddles and ward boards. It was helpful to know that these would be useful additions (page 15, lines 322-324, page 16, line 336).

7) In my view I would be cautious about identifying the paper as the first in a vast subject such as teamwork. Staff views, in the context of hip fracture, the current culture and at this particular time may be key. It is suggested that the model will be useful outside of the UK which does not take into account context and culture. I would be inclined to revisit the strengths and limitations section below the abstract and check that there is enough information to support these in the text.

Thank you, we have edited throughout the manuscript to emphasise that the focus of our work is hip fracture. We retain some of the broader literature as a way of positioning the work and to provide relevance for those interested in other fields. some of our hyperbole and we no longer refer to the potential international relevance of the work. We have also revisited the strengths and limitations section below the abstract and made changes so that it more clearly states key aspects of the study and does not make overstatement. Instead, we have explained that this work is needed in relation to hip fracture care because of the multidisciplinary nature of such

care, so that our study may help to inform care in contexts in which multidisciplinary might also be relevant.

8) Abstract, I would identify all the types of staff you included.

We would very much like to have included this information, and we have added additional text to respond to other comments, and we remain at the word limit of 300 words. The information is contained in the participants and results sections, which we hope will be sufficient, the relevant information can now be found on lines 269-277 on page 12 and in Table 1.

9) Did anything useful evolve from the PPI groups that influenced any part of the study?

Thank you, it was interesting to work with a patient group in relation to a study that was carried out with healthcare professionals as participants but that was of course focused on patient care. We have added detail to the PPI section to say briefly that members' input helped to shape study documentation, and they felt that findings resonated with their experiences, especially their concerns that patient information was sometimes not well communicated within healthcare (page 12, lines 259-263).

10) All the best for your ongoing work.

Thank you, we very much appreciated your thorough, professional and kind peer review.

Reviewer: 2

Competing interests of Reviewer: None

VERSION 2 – REVIEW

REVIEWER	Snowdon, David Monash University, Peninsula Clinical School
REVIEW RETURNED	07-Aug-2023

GENERAL COMMENTS	Thank you for addressing my comments.
---------------------------------------

REVIEWER	Tutton, Elizabeth University of Oxford, NDORMS
REVIEW RETURNED	04-Aug-2023

GENERAL COMMENTS	Thank you for your thoughtful and detailed consideration of the reviewers' comments. A really helpful paper that I enjoyed reading. Strong messages regarding service improvement. Could you check the last but one quote (Lance), it wasn't quite clear to me re 'opinion of the nurse'.
---

VERSION 2 – AUTHOR RESPONSE

Reviewer: 2 Dr. Elizabeth Tutton, University of Oxford

Comments to the Author:

Thank you for your thoughtful and detailed consideration of the reviewers' comments. A really helpful paper that I enjoyed reading. Strong messages regarding service improvement. Could you check the last but one quote (Lance), it wasn't quite clear to me re 'opinion of the nurse'.

Thank you, we are pleased that you found our response to reviewers' comments helpful and we appreciated your suggestions. This quotation relates to the importance of recognising the value of different professional groups. We agree that it's meaning is unclear and have edited it using brackets to provide clarity (Table 3).